# Influence of Mechanical Couplings on the Dynamical Behavior and Energy Harvesting of a Composite Structure

**DOI:** 10.3390/polym13010066

**Published:** 2020-12-26

**Authors:** Marek Borowiec, Jaroslaw Gawryluk, Marcin Bochenski

**Affiliations:** Department of Applied Mechanics, Lublin University of Technology, Nadbystrzycka 36, 20-618 Lublin, Poland; j.gawryluk@pollub.pl (J.G.); m.bochenski@pollub.pl (M.B.)

**Keywords:** sequences of plies, composite beam, energy harvester, MFC element, FEM dynamic analysis

## Abstract

In this paper, the dynamical behavior of composite material is analyzed, including the energy harvesting effect. The composite is modeled by the Finite Element Method (FEM) and is made of pre-impregnate with a matrix of thermosetting epoxy resin reinforced with high-strength R-type glass fibers, and it is designed as a beam structure that is exposed to mechanical vibrations. The structure assumed the form of a beam with a substantially rectangular cross section. The couplings of motion occurring between mode shapes at properly selected fiber orientations are investigated. The beams with determined sets of composite layers and a coupling effect are used to recover electricity from the mechanical vibrations in the vicinity of the first resonance zone. The composite with a certain number of fiber glass layers has assumed an orientation relative to the beam axis. The new values found in this paper are the intensity of the coupling between the bending in the stiff and flexible directions of the beam for a chosen fiber layer stacking sequence. Additionally, the influence of layer configuration on the energy harvesting efficiency of the Macro-Fiber Composite (MFC) piezoelectric element is assessed.

## 1. Introduction

Wires, shafts, beams, and shells are basic components that are widely used in structural engineering. Recent advances in sophisticated analytical and numerical models allow us to predict the nonlinear dynamics of systems when the frequency of excitation is very close to natural frequencies. In our study, we are limited only to beams, which have a very rich dynamic behavior depending on boundary conditions. For instance, numerical Finite Element (FE) simulations on clamped–clamped and hinged–hinged planar beams with homogeneous material properties show a hardening nature, while simply supported boundary conditions cause the softening of the frequency response for the first flexural mode [1]. This phenomenon has been investigated analytically and numerically in planar beams by means of axial–transversal coupling in nonlinear frequency amplitude response via implementing an additional axial spring [2], lumped tip mass [3], and higher-order resonances [4]. The predicted behaviors were numerically (FEM) and experimentally validated on a kinematically excited prototype with an additional mass moment of inertia caused by physical hinges [5]. In addition, the hardening–softening dichotomy, transversal–transversal (one to three), transversal–longitudinal (one to two) internal resonances, and detached solutions were detected [2,6]. All of the aforesaid complexities impede time-consuming finite element simulations. Moreover, bifurcation solutions cannot be achieved in the presented Abaqus_CAE dynamic implicit solver. Avoiding problems with the time-consuming integration and potential of the design setup, we limit our case study only to the clamped-free beam, in which frequencies corresponding to longitudinal modes are well separated from frequencies associated with the first orthogonal transversal resonances. The primary resonances of cantilever beams are almost independent of the amplitude of excitation, namely, their frequency responses have linear characteristics [7]. Moreover, the amplitude response to the amplitude of the excitation ratio has the highest value and this benefit is desirable in relation to energy harvesting issues. Dynamical coupling between two orthogonal modes, e.g., flexural–flexural, is not applicable for isotropic material, but can be achieved with commonly used unidirectional composite materials. The configuration of plies in composite structures plays an important role in their dynamic properties. In the design process of complex composite materials, a sequence of laminate layers can have a significant influence on the mechanical properties of a structure and thus on its dynamic behavior [8]. This applies to thin-walled composite structures, in which—through the appropriate sequence of layers—it is possible to achieve interesting coupling effects [9]. The aforementioned effects between different vibration modes is particularly interesting when taking into account the study of dynamic structures [10]. Santiuste et al. [11] studied the dynamic behavior of rectangular cross section laminated beams, considering the bending–torsion coupling effect. An analytical model is studied, taking into account the Flexibility Influence Functions Method and the results are compared with a three-dimensional numerical model. It has been concluded that the bending–torsion coupling effect needs to be considered in a one-dimensional model because this coupling can have a considerable effect on laminates with lamina orientations other than 0° and 90° degrees. Gawryluk et al. [12] conducted a numerical and experimental modal analysis of laminated thin-walled beams, where a flapwise bending with torsion coupling effect or a flapwise–chordwise bending coupling effect took place. Circumferentially asymmetric stiffness and circumferentially uniform stiffness beams are analyzed. The numerical simulations are carried out and some cases are validated experimentally. Additionally, the influence of the fiber orientation in laminates on the torsion stiffeners of these beams is discussed. It has been determined that the lateral and transversal displacements of the cross section for the circumferentially uniform stiffness beam strongly depend on the fiber orientation. Latalski et al. [13] studied the coupled flexural–flexural vibrations of a rotating composite beam with an MFC element. The transversal and lateral bending modal couplings are obtained through the use of the directional properties of the beam’s laminate and ply stacking distribution. A mathematical model of a rotating rigid hub with a flexible composite beam has been discussed by Latalski et al. [14]. The authors considered differential equations of motion featuring beam bend–twist elastic coupling. Additionally, parametric studies are conducted to evaluate beam stiffness coefficients depending on the fiber lamination angle. Czapski et al. [15] studied the influence of the angular arrangement of laminae on the buckling force of rectangular composite plates. A general, simple analytical method of buckling load determination and finding the best arrangement in terms of the highest stiffness in the pre-buckling state are presented. The results obtained for characteristic layups are compared with FEM and the literature results. The coupling effects of bending and twisting, as well as the buckling load of coupled structures, are verified in [16]. The analytical expressions for the torsional distortion of the bending–twisting coupled structures are derived. The authors used the Differential Evolution algorithm to optimize the bending–twisting coupled structures for the purpose of maximizing torsional distortion. It has been concluded by the authors of [16] that introducing the bending–twisting coupling effect of laminates into the optimal design of structures can significantly improve their bending–twisting coupling effect. This also improves the buckling load of the majority of structures, but reduces their yield strength to some extent. Finite element models (FEMs) are used in the design of composite bend–twist (BT) coupled structures. In [17], the experimental verification of composite finite element models by static bending tests of bend–twist coupled laminate plates is presented. It has been found that the bending and twisting responses of the structures are linear and within the range of applied loads. However, the laminate thickness and accuracy of ply angles are highly influential on the ability of the FEM to predict accurate deformations. Teter et al. [18] demonstrated the numerical and experimental analysis of a thin-walled composite box-beams of two cases—one undamaged and another damaged. Two types of beam configuration are examined: the first is characterized by the coupling between bending in a flexible direction and torsion, while the second configuration is characterized by coupled bending in two perpendicular directions (flexible and stiff). It has been concluded by Teter et al. [18] that damage to the structure, such as cracks, led to the rotation of the cross section. Falkowicz et al. [19] examined the effects of lamina ply orientation on the critical state and behavior of thin-walled composite plates with a cut-out in the post-critical range under compression. It has been found that the angle of ply orientation in laminate has a significant effect on the mechanical properties of elastic plate elements with a central cut-out. The influence of the structural geometry (i.e., the sequence of laminate layers, different materials, and their stiffness) on the electromechanical coupling occurring in energy harvesters is very interesting. Energy harvesting from mechanical vibration, biological systems, and the effects of power harvesting on the vibration of a structure are presented in three review papers [20,21,22]. The influence of the geometric and material properties of core and metallic layers on the performance of the sandwich piezoelectric energy harvester are analyzed in [23]. For such models, the modal analysis is useful to reveal the composite characteristics [24], as it is provided in [25] that the material properties strongly influence the dynamical behavior. The mathematical formulation of a generalized electromechanical model of this structure is developed using the Lagrange approach. The analytical modeling is validated by finite element simulations and experimental tests. Borowiec et al. [26] used the Macro-Fiber Composite (MFC) element to harvest energy from the vibrations of composite structures. The effects of the system load on the generated voltage in an active element are tested experimentally and numerically. In [27], a vibration energy harvesting system prototype with MFC patches bonded to a cantilever beam is presented. A finite element model is established to estimate the output voltage of the MFC harvester. The numerical results are validated by the experimental ones. The authors attempted to optimize the efficiency of energy harvesting and the geometric configurations of the cantilever beam, as well as the effects of the electrical properties of the MCF element. The aim of this paper was to design a finite element beam MFC model in Abaqus_CAE software, which is capable of harvesting energy from mechanical vibrations and describing the influence of the fiber configurations in composite beam systems on mechanical bending–bending couplings. The dynamical behaviors of the mechanically coupled resonator was tested on the energy harvesting efficiency due to both electromechanical (piezoelement–substructure) and geometrical (fiber arrangement) couplings. The dynamical characteristics of composite beams with significantly rectangular cross sections were determined to reveal any nonlinear behaviors. The conditions of the appearance of the bending–bending coupling effects were investigated, which assured the highest possible level of response vibration in the flexible direction, while the system was excited in the stiff direction. It was demonstrated that the discovered characteristics of the chosen fibers arrangements have a nonlinear character. Moreover, the voltages generated on the piezoelement were estimated for energy harvesting in analyzed conditions. The remainder of this paper is organized as follows. In Section 2, the numerical approach is described; then, the results of the demanding simulations are discussed in Section 3. Finally, the results are concluded in Section 4.

## 2. Composite Active Beam Modeling

The numerical analyses are conducted by the finite element method (FEM) in Abaqus software. The composite beam is modeled as a continuum shell finite element, SC8R. There are 8-node 2nd-order continuum elements with square-shaped functions and three translational degrees of freedom at each node with reduced integration. These elements also have rotational degrees of freedom. The sequence of the laminate layers is defined using the layup-ply technique. Each layer is made of a glass–epoxy unidirectional prepreg.

The Macro-Fiber Composite element is modeled as a C3D20RE solid element. In the numerical model, only the active part of the MFC element (85 × 14 mm) is considered. One end of the composite beam is fixed (Figure 1a). Additionally, at one end of the MFC, a zero- voltage signal is set. It is realized as an electric boundary condition. Between the beam and the active element, “TIE” interactions are defined. The geometrical dimensions of the applied elements are presented in Table 1 and the parameters of the laminate and the active element are provided in Table 2. The values of used parameters have been verified in paper [28]. The displacements of the beam are realized in two perpendicular directions, as shown in Figure 1b. Such dynamical behavior, manifested by vibrations in both directions, is possible because mechanical coupling occurred. Due to the long simulation time (1 point in the amplitude–frequency diagram corresponds to 4 h of simulation time), it was necessary to carefully plan the numerical research. Hence, in the numerical models, we assume three angular positions for the layers: 30°, 45°, and 60° in two configurations (symmetric and antisymmetric), in relation to the middle layer at 0°.

The six geometrical sets of composite beams are investigated in detail. The thickness of each layer in the analyzed composite beam was assumed to be 0.25 mm, and the beams consisted of eleven layers. The numerical dynamic studies, with kinematic excitation defined as a mechanical boundary condition, are performed. This excitation is characterized by the locked displacements of both of the nodes in the longitudinal U3(X,Y,Z) and flexible U2(X,Y,Z) directions of one beam end. However, in the stiff direction U1(X,Y,Z), the periodic beam motion is described by the equation $u_{x}=A_{exc}\sin\left(\omega t\right)$, where $A_{exc}$ is an excitation amplitude ranging from 1 mm to 20 mm, and ω is the circular frequency of excitation. The implicit procedure in Abaqus software is used to simulate the time domain response of the structure. The main result the coupling of displacements in two perpendicular directions—stiff and flexible—as well as the voltage output of the consecutive beams during vibration. It is expected that the energy harvesting will differ when taking into account the different beam behaviors.

## 3. Numerical Results

In this section, the numerical results from the Abaqus simulations are reported. The composite beams are divided into sections for the various directions of the layers. Due to layer configurations, the beams reached various stiffnesses, which is confirmed by the natural frequencies of the analyzed stacking sequence sets. In Figure 2, we plot the time series at resonance excitation frequencies, according to the values listed in Table 3 at an excitation amplitude of $A_{exc}=1$ mm. The beam responses clearly differed for each of the set angles, as well as regarding the symmetric and antisymmetric configurations of the fiber layers in the frame of the same angle set. The stacking sequence is the factor that has the most influence on the beam behavior at configuration +30°/−30° (Figure 2a) but its effect resolutely disappeared in the case of 45° and 60° (Figure 2b,c). Therefore, the beam outputs were simulated in the vicinity of the widened resonance zone ($0.5f_{n}$–$1.5f_{n}$) at the same excitation amplitude $A_{exc}=1$ mm for the investigation of any nonlinearities in the response behaviors.

This excitation resonance range was 10–25 Hz; the response of the amplitude–frequency characteristics of beams are presented in Figure 3a–c. Due to the different stacking sequences of the fiber layers, it is clear that the resonance peaks moved to the right of the configuration sets (+/−) in relation to sets (+/+). This reveals that the stiffness of composite beams is increasing for the layer configuration sets (+/−). According to the time series in Figure 2, the case of the antisymmetric configuration +30°/−30° response was the most sensitive to the excitation in the stiff direction, while a variant of the output of configuration +60°/−60° shows that the displacement amplitude disappeared.

Moreover, in the case of angle configuration 30°, the significant amplitude response of the system in the resonance zone for the antisymmetrical stacking sequence (+/−) is more than two times higher than the symmetrical one (+/+). The opposite situation occurs in the case of angle configuration 60°, where the amplitude response for the symmetrical stacking sequence (+/+) is significantly higher (six times) than the antisymmetrical stacking sequence (+/−) (see Figure 3c). However, in both symmetric and antisymmetric configurations, as represented by the case of angle configuration 45°, comparable amplitudes are achieved.

While the angle value decreases, the deviation of the natural frequency is visible for both configurations (+/−) and (+/+). For the 30° angle they are 19.89 Hz and 18.31 Hz, respectively (see Table 3). In all analyzed cases, an evident pick in the frequency–amplitude plot is observed in the vicinity of the first resonance zone, with the exception of the antisymmetric 60° case, where the amplitude–frequency curve appears to be flat.

Based on the displacements of points 1 and 2 (Figure 1b), the rotation of the free end of the beam has been determined. For antisymmetric stacking sequence configuration (+/−), this twist effect is close to zero regardless of the angle values of the composite fibers (see blue lines in Figure 4a–c). However, for symmetric stacking sequence configuration (+/+), the greater the twist of the beam end, the smaller the angle of the fibers. This phenomenon revealed that a slight tendency towards the twisting effect is more likely in any symmetric configurations. The next observation was provided for an increased excitation amplitude at at a resonance excitation frequency.

By studying the results presented in Figure 5a–c, one can note that, for the antisymmetric stacking sequence configuration (+/−) of 30° and 45°, the amplitude response of the beam depends linearly on the excitation amplitude. The nonlinear (regressive) character of the 30° and 45° angle sets is clearly visible for the symmetric configuration (+/+) (Figure 5b,c). The opposite behavior was observed in the case of the 60° angle set presented in Figure 5d. For the symmetric configuration (+/+) the characteristic is close to a linear one, but for the antisymmetric one (+/−), the nonlinear progressive characteristic is observed. Additionally, while excitation amplitude was increasing, the phenomenon of the movement of the vibration center was observed (Figure 6a,b).

This movement reveals the progressive increase in relation to the excitation amplitude. Moreover, the slight intensification of this phenomenon occurs for the 30° and 60° sets of angles and for symmetric (+/+) configurations.

The above results revealed that one can expect higher levels of vibrations in the flexible direction with an antisymmetrical stacking sequence (+/−) than with a symmetrical one (+/+), whereas excitation operates in the opposite manner in the stiff direction. Therefore, the influence of the stacking sequence angle in the antisymmetric case of the beam’s response in the flexible direction has been checked more precisely and reported in Figure 7a. The excitation amplitude was fixed at 1mm, while the angle was increased from 5° to 85° with a 5° increment. The numerical investigation showed that the bending–bending coupling from small angles grew rapidly to reach a maximum at 15°, (with there being a maximum vibration amplitude for the beam in the flexible direction); afterwards, it is gradually decreased, reaching a value close to zero at 60° and, again, growing slightly, reaching a local maximum at 75°. Such behavior reveals the nonlinear characteristic of the analyzed composite beam. One can notice that, in the case of the beam with a 75° stacking sequence, the response of the beam reaches 10 % of the maximum amplitude found at 15°. In Figure 7b, the root mean square (RMS) of the voltage generated on the piezo-element is plotted. These responses have characteristics similar to the output amplitude of the beam’s displacement in the flexible direction (see Figure 7a). One can conclude that the harvested energy strongly depends on the mechanical coupling resulting from the stacking sequences of layers.

In Figure 8a–d, the selected time series of the voltages are presented. Note that, simultaneously, the angle sets of fibers, the mutual orientation of layers (symmetric and antisymmetric) and the amplitude of excitation have strong influences on the shapes of voltage signals generated on the MFC. For small excitation amplitudes $A_{exc}=1$ mm, voltage oscillates nearly symmetrically around zero. While the excitation amplitude is significantly greater ($A_{exc}=20$ mm), the voltage oscillates around certain constant values (DC components). In some cases of the angle configurations, the center point of the voltage series offset significantly exceeds its amplitude, i.e., +60°/−60° (Figure 8c). Based on the generated time series of the voltage, the RMS and the offset of the center point for symmetric and antisymmetric configurations and angle sets 30°, 45°, and 60° are plotted in Figure 9a,b. The highest RMS value is observed for the 30° antisymmetric configuration and its characteristic is almost linear. In other cases, slightly progressive characteristics are observed. The next configuration in respect to the obtained voltage generates RMS values more than twice as low (30° symmetric configuration). The differences between symmetrical and antisymmetrical configurations for 45° and 60° angles are much smaller. It can be concluded that the RMS value decreases with the angle of the fibers and that it is always greater for antisymmetric configurations than for symmetrical ones. However, when analyzing Figure 7b, for 60° there is a minimum voltage and, after exceeding this value of the angle, a slightly increasing RMS value is observed. The results presented in Figure 9b show that the DC voltage component moves progressively with an increasing excitation amplitude. Moreover, for the 30° and 45° configurations, this value does not depend much on the type of configuration (symmetrical or antisymmetrical). The greater effect is seen at an angle of 60°. Due to the capacitive nature of the current source of Macro Fiber Composites (MFCs), while a resistive load is applied, the DC voltage component will be shifted to zero [31]. This DC component has no significant energy in comparison to the AC component.

## 4. Conclusions

In this paper, the influence of layer configuration on the dynamical behavior of a structure is reported. This leads to significant differences in the effectiveness of energy harvesting on MFC elements. The most important features of the proposed structure are described. The antisymmetrical configurations are characterized by a higher stiffness than the symmetrical ones, regardless of the fiber angles. Bending–torsional coupling was observed only in the symmetrical configurations. Among the cases considered in detail, the greatest bending–bending couplings were observed for the 30° angle in the antisymmetric configuration. However, the analysis performed in a wider range of fiber angles at the excitation amplitude $A_{exc}=1$ mm indicates that the maximum value of this coupling may occur around 15°. Following the characteristics of the beam vibration amplitude in the flexible direction versus the excitation amplitude in the stiff direction, a clear nonlinearity of a regressive nature for symmetrical configurations of 30° and 45° is found. Moreover, a nonlinearity of a progressive nature for the antisymmetrical configuration of 60° is also observed. This denotes that the analyzed system manifests complex dynamics. In all numerically analyzed cases, the movement of the vibration center is observed, which is strongly dependent on the excitation amplitude. Structural analysis, in the context of dynamical behavior, is complex due to the fiber configurations of composite plies. The main advantage of the paper is found in the mechanical behavior between the perpendicular directions of the anisotropic laminated composite material during vibrations in a wide range of excitation frequencies and amplitudes. The next step of the project will be the experimental analysis and extension of the numerical models for the investigation of the strongly nonlinear effects expected at the modified boundary conditions of the systems.

## Figures and Tables

**Figure 1 polymers-13-00066-f001:**
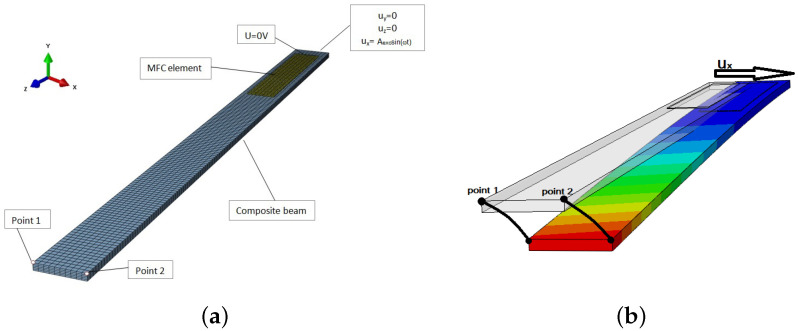
Finite Element (FEM) model of the beam with the piezoelectric element (**a**); beam deformations in flexible direction by excitation in stiff direction $u_{x}$ at chosen configuration of layers (**b**).

**Figure 2 polymers-13-00066-f002:**
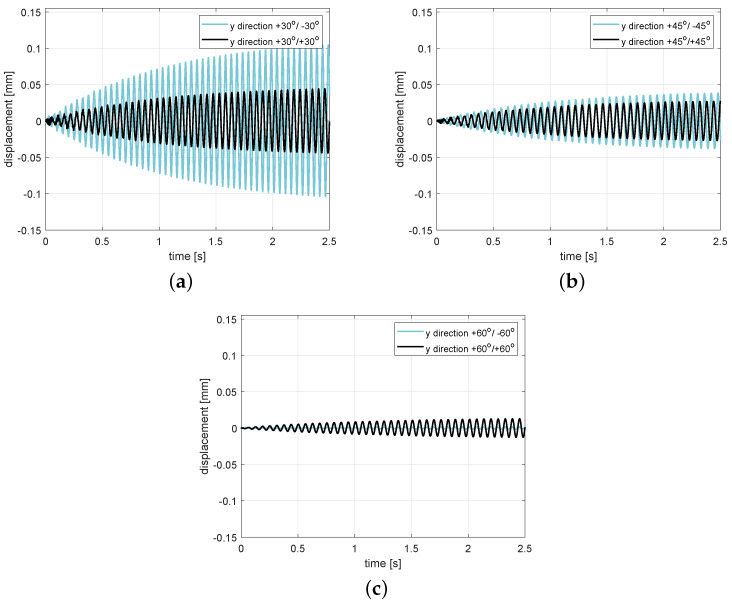
Magnified response of the composite beam after transient time at resonance frequency excitation with excitation amplitude $A_{exc}=1$ mm for 30° (**a**), 45° (**b**), and (**c**) 60° layer stacking sequences, respectively. (Details in [30]).

**Figure 3 polymers-13-00066-f003:**
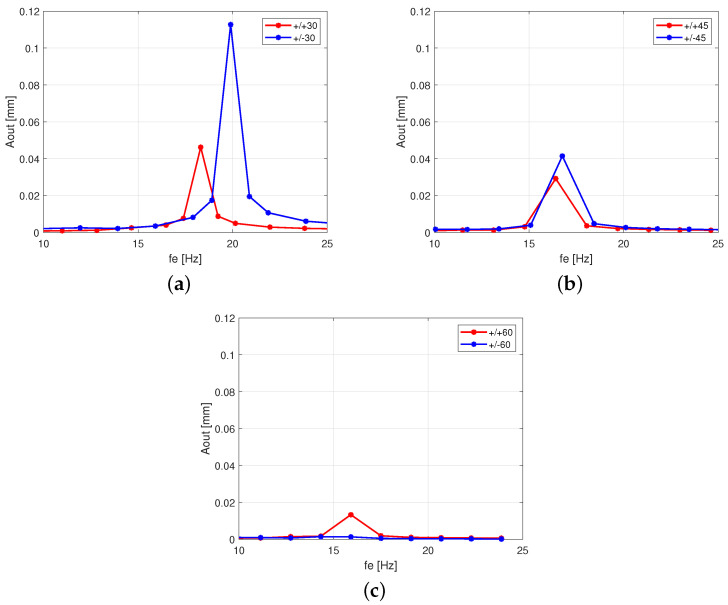
Response amplitude versus excitation frequency $f_{e}$ at $A_{exc}$ = 1 mm for fiber orientations 30° (**a**), 45° (**b**), and (**c**) 60°, respectively.

**Figure 4 polymers-13-00066-f004:**
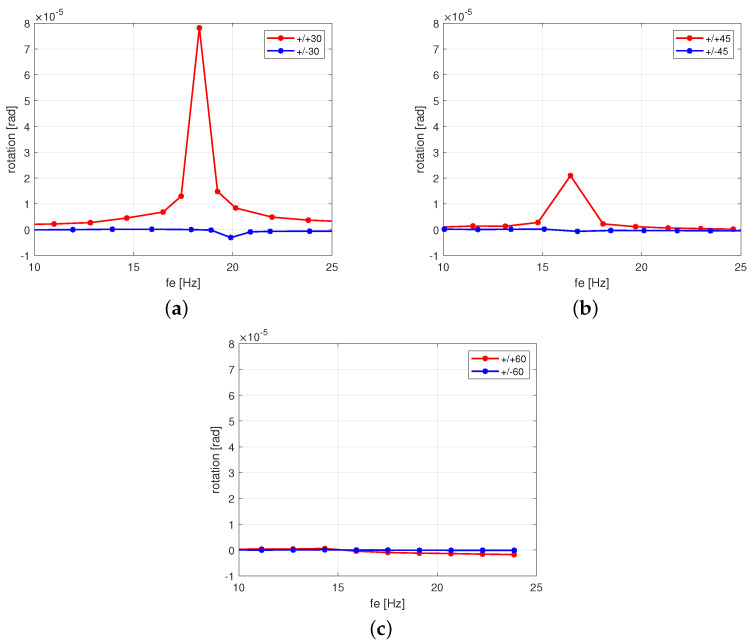
Response of rotation versus excitation frequency $f_{e}$ at $A_{exc}$ = 1 mm for fiber orientations 30° (**a**), 45° (**b**), and (**c**) 60°, respectively.

**Figure 5 polymers-13-00066-f005:**
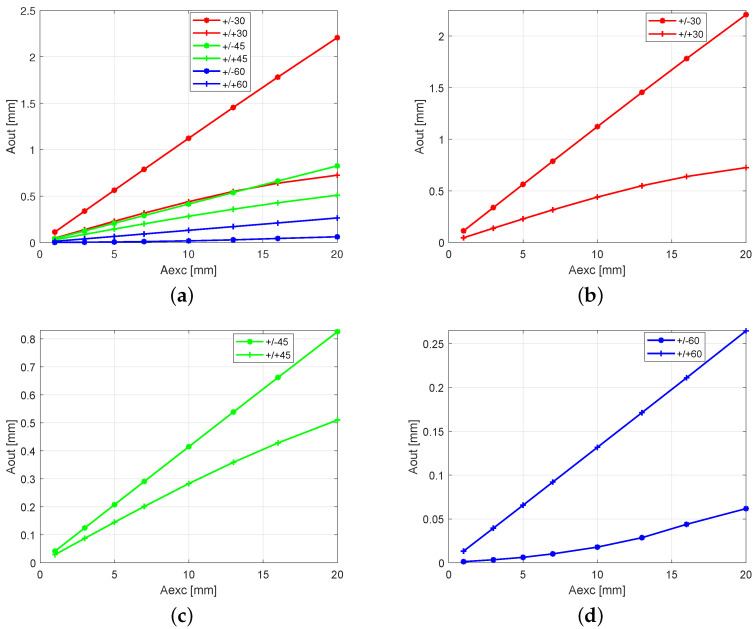
Response amplitude for all analyzed cases of fiber orientations against the increased excitation amplitude: (**a**) $A_{exc}$ = 1 mm, 3 mm, 5 mm, 7 mm, 10 mm, 13 mm, 16 mm, and 20 mm, respectively, and (**b**–**d**) the magnified results of the three fiber configuration cases.

**Figure 6 polymers-13-00066-f006:**
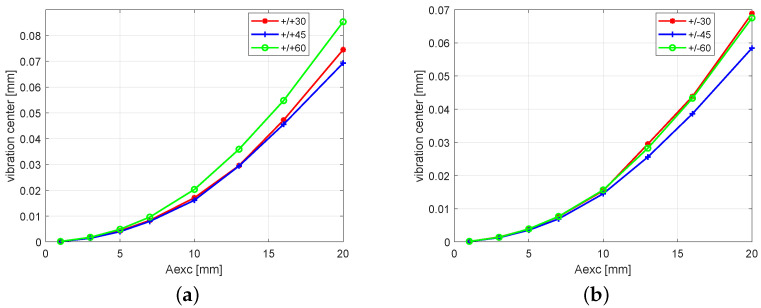
Movement of the vibration center depending on the fiber set and the excitation amplitude for symmetric (**a**), and antisymmetric configurations (**b**), respectively.

**Figure 7 polymers-13-00066-f007:**
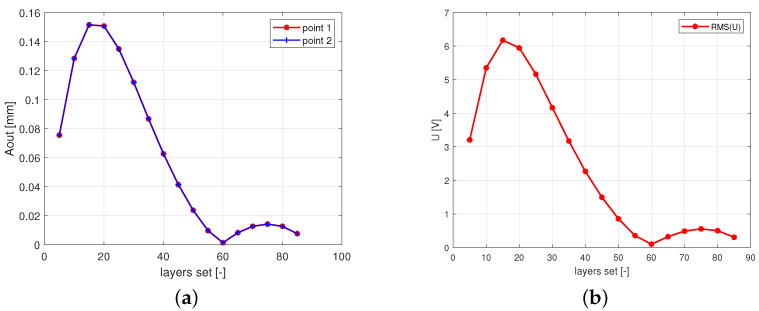
Amplitude response of the antisymmetric beam (**a**), and the root mean square of the electric potential on the piezoelectric patches via fiber configuration sequences (**b**).

**Figure 8 polymers-13-00066-f008:**
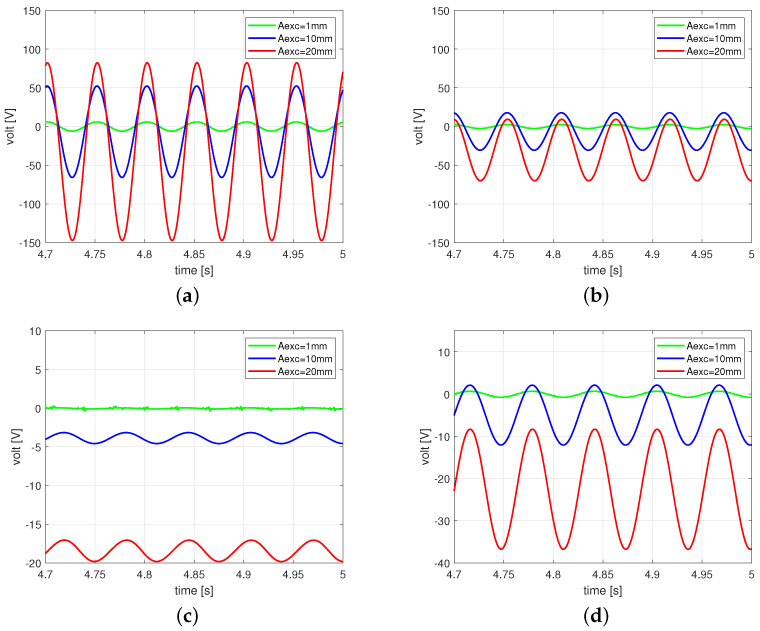
Voltage time series at three selected excitation amplitudes for (**a**) +30°/−30° set, (**b**) +30°/+30°, (**c**) +60°/−60°, and (**d**) +60°/+60°, respectively.

**Figure 9 polymers-13-00066-f009:**
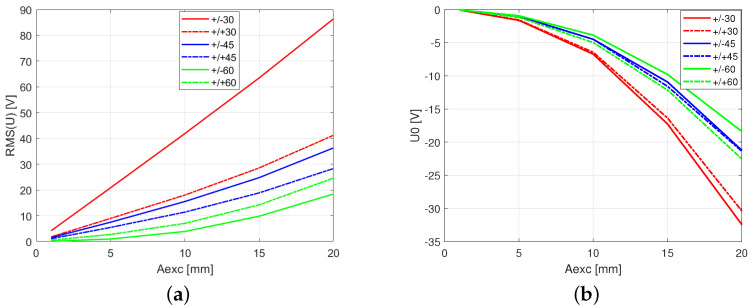
Root mean square of voltage response (**a**,**b**) the shift in the mean value of the voltage potential, calculated according to the formula $U0=0.5(U_{max}+U_{min})$.

**Table 1 polymers-13-00066-t001:** Dimensions of the composite beam and applied element of piezo from Smart Material [29].

Dimensions	Beam	Piezo Type M8514-P1
length	300 mm	85 mm
width	20 mm	14 mm
thickness	2.75 mm	0.3 mm

**Table 2 polymers-13-00066-t002:** The parameters of the structure.

**Composite Layer**	
longitudinal modulus	46.4 GPa
transverse modulus	14.9 GPa
shear modulus	5.20 GPa
Poisson’s ratio	0.27
mass density	2032 kg/m^3^
**Active Element**	
Young’s modulus	6.75 GPa
Poisson’s ratio	0.31
piezoelectric constant	1.02 × $10^{-7}$ m/V

**Table 3 polymers-13-00066-t003:** The natural frequencies of the composite beam.

Stacking Sequence	$f_{n}$ (Hz)
+30° (5)/0/−30° (5)	19.89
+30° (5)/0/+30° (5)	18.31
+45° (5)/0/−45° (5)	16.76
+45° (5)/0/+45° (5)	16.41
+60° (5)/0/−60° (5)	15.92
+60° (5)/0/+60° (5)	15.91

## Data Availability

All the experimental data herein presented are made available up-on request to the corresponding author.

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
