# Peer review of "Influence of Mechanical Couplings on the Dynamical Behavior and Energy Harvesting of a Composite Structure"

_polymers, 2020, doi:10.3390/polym13010066_

Round 1
Reviewer 1 Report
Influence of the Mechanical Couplings on Dynamical Behavior and Energy Harvesting in Composite Structure
My comments are:
- It is better author should highlight some major outcomes in the abstract.
- Abstract is good, but lack of applications related to the presented study. I suggest Author should divide the introduction into Background, literature survey, and the proposed work. Etc.
- Provide the source for table 1.
- Author should add some mathematical formulas to make the outcomes more plausible.
- If possible then author must add a comparison to make the result interesting.
- In the conclusion, please show how the work advances the field from the present state of knowledge. Please provide a clear justification for your work in this section, and indicate uses and extensions if appropriate. Moreover, you can suggest future experiments/simulations and point out those that are underway.
- English should be improved, minor glitches must be removed.
Reviewer 2 Report
In this manuscript “Influence of the Mechanical Couplings on Dynamical Behavior and Energy Harvesting in Composite Structure”, is analyzed the dynamical behavior and the energy harvesting effect on the composite structure made of epoxy resin reinforced with high-strength glass fibers. The structure is in form of a beam with a rectangular cross-section. The authors reported the influence of layers configuration on the dynamical behavior of the structure. The anti-symmetrical configuration are characterized by a stiffness higher than the symmetrical one. I consider that the manuscript requires an improvement in the following sections:
- Is necessary to indicate the main contribution of this study.
- Composite molding. Provide a brief description of the preparation of the laminate material, not just refer to another article.
- This section describes the results, they are not analyzed (only one article is cited).
Reviewer 3 Report
Very interesting topic.Sufficient originality. However the paper is flawed due to the following main issues:
1.The investigation is only theoretical without comparison with experimental results.
2.The references should be updated.
3.The conclusion is too brief.
4.English should be improved
